# MAKING RETRIEVAL-AUGMENTED LANGUAGE MODELS ROBUST TO IRRELEVANT CONTEXT

**Ori Yoran**[1]  **Tomer Wolfson**[1,2]  **Ori Ram**[1]  **Jonathan Berant**[1]
[1]Tel Aviv University, [2]Allen Institute for AI
{ori.yoran, ori.ram, joberant}@cs.tau.ac.il  tomerw@allenai.org

## ABSTRACT

Retrieval-augmented language models (RALMs) hold promise to produce language understanding systems that are are factual, efficient, and up-to-date. An important desideratum of RALMs, is that retrieved information helps model performance when it is relevant, and does not harm performance when it is not. This is particularly important in multi-hop reasoning scenarios, where misuse of irrelevant evidence can lead to cascading errors. However, recent work has shown that retrieval augmentation can sometimes have a negative effect on performance. In this work, we present a thorough analysis on five open-domain question answering benchmarks, characterizing cases when retrieval reduces accuracy. We then propose two methods to mitigate this issue. First, a simple baseline that filters out retrieved passages that do not entail question-answer pairs according to a natural language inference (NLI) model. This is effective in preventing performance reduction, but at a cost of also discarding relevant passages. Thus, we propose a method for automatically generating data to fine-tune the language model to properly leverage retrieved passages, including for challenging multi-hop tasks, using a mix of relevant and irrelevant contexts at training time. We empirically show that even 1,000 examples suffice to train the model to be robust to irrelevant contexts while maintaining high performance on examples with relevant ones.

## 1 INTRODUCTION

Large Language Models (LLMs) (Brown et al., 2020; Chowdhery et al., 2022; Touvron et al., 2023) are the foundation on top of which modern language systems are built. However, open-domain question answering (ODQA; Chen et al. 2017) and other knowledge-intensive tasks (Thorne et al., 2018; Petroni et al., 2021) require vast amounts of up-to-date factual knowledge about rare entities that even very large models cannot memorize (Roberts et al., 2020; Dhingra et al., 2022). A dominant approach for combating this issue has been Retrieval Augmented Language Models (RALMs), which incorporate a retrieval mechanism to reduce the need for storing information in the LLM parameters (Guu et al., 2020; Lewis et al., 2020b; Izacard et al., 2023; Rubin & Berant, 2023). Furthermore, RALMs have also been shown to improve ODQA performance in an in-context setting (without any training), simply by prepending retrieved sentences to the input question (Ram et al., 2023). Nevertheless, retrievers are not perfect and past work has shown that noisy retrieval can negatively affect LLM performance (Petroni et al., 2020; Li et al., 2023). For example, in Fig. 1, when posed with the questions *"Who is playing Jason on General Hospital?"* a vanilla LLM (left) correctly answers the question while the RALM (right) is "distracted" by irrelevant context about the actor portraying Cooper, not Jason.

In this work, we analyze and improve the robustness of RALMs to noisy retrieved contexts. Our definition for *retrieval-robust LLMs* states that: (a) when relevant, the retrieved context should improve model performance; (b) when irrelevant, the retrieved context should not hurt model performance. To this end, we present two methods for retrieval-robustness in RALMs (§2).

First, we consider a setting where we have black-box access to the LLM and cannot train it. Rather than solely relying on in-context prompting (Brown et al., 2020), we frame retrieval robustness as a natural language inference (NLI) problem (Dagan et al., 2006; Bowman et al., 2015). Namely, given a question and retrieved context, an NLI model can predict whether a question-answer pair

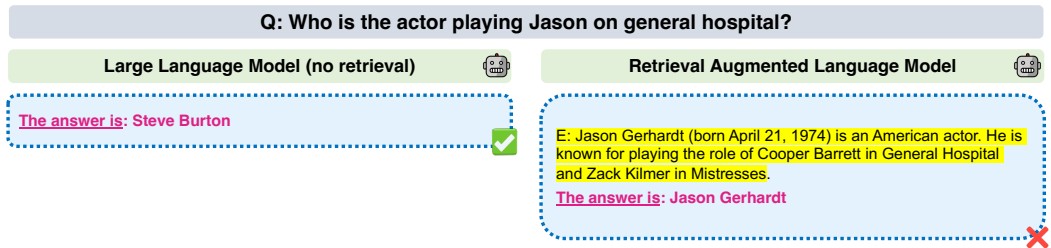

Figure 1: An example from NQ where retrieval augmentation causes *Llama-2-13B* to err. Augmenting with irrelevant retrieved context leads to an error (right), although the model is able to answer the question without retrieval (left).

(hypothesis) is entailed by the context (premise). Building on the strong performance of recent NLI models (e.g., in detecting model hallucinations (Honovich et al., 2022) and attributed question answering (Bohnet et al., 2023)), we use such models to identify irrelevant contexts. When the context is labeled as irrelevant to the question-answer pair, we generate the answer using the LLM *without retrieval* as a "back-off strategy". Our results show that this natural baseline is highly effective at identifying irrelevant contexts, but is too strict and discards relevant ones as well (§4).

We then propose a method for training RALMs to be retrieval-robust. Intuitively, LLMs are not trained with retrieved passages, and thus brittleness to noisy retrieval is somewhat expected. Therefore, we perform an additional finetuning step that teaches the LLM to be robust to noisy contexts. The core challenge is to generate data for finetuning, and we describe a procedure for automatically generating such data for both single-hop and multi-hop questions. In the single-hop setting, assuming access to gold QA pairs and a retriever, we create training examples using retrieved contexts, where we can use low-ranked or random passages as noisy contexts. In the multi-hop setting, training examples need to contain not only retrieved contexts, but also intermediate questions, answers and relevant contexts, which comprise the *question decomposition* (Fig. 3), shown to be necessary for high performance on multi-hop questions (Wolfson et al., 2020; Press et al., 2023). To generate decompositions to train on, we use a strong LLM, prompted for decomposition without any retrieval. Then, we can sample multiple decompositions, and use self-consistency (Wang et al., 2023) to identify high-quality training examples (§3.2.3).

To test our methods, we evaluate retrieval robustness on five ODQA benchmarks, four of which contain multi-hop questions, where the retriever is called multiple times (Jiang et al., 2023). Fig. 2 shows that even with a strong retriever (top-1 Google search) incorporating the retrieved context actually *hurts* model performance on two of the benchmarks (STRATEGYQA and FERMI). Moreover, adding randomly-retrieved contexts dramatically decreases accuracy on all five datasets. Our analysis (§5) shows that irrelevant context causes a wide range of errors, which include copying irrelevant answers from the retrieved sentences and hallucinating incorrect answers and decompositions.

Our results demonstrate that finetuning LLMs to be retrieval-robust enables them to ignore irrelevant context while improving their overall accuracy (§4). When using a strong retriever at test time, our finetuned models outperform both models that were finetuned without retrieval, as well as untrained models prompted using in-context learning. To test robustness to *noisy context*, we evaluate QA accuracy when models are given randomly-retrieved contexts. In this setting, our finetuned models perform on par with those that were finetuned *without* retrieval, demonstrating retrieval robustness. In addition, our ablation study shows that training models on a mixture of relevant and irrelevant contexts results in models that are much more robust to irrelevant context.

To summarize, our main contributions are:

- We conduct a thorough analysis on the robustness of RALMs to irrelevant retrieved contexts.
- We show that small NLI models can be used to identify irrelevant context and improve robustness, without updating the model parameters.
- We demonstrate that training LLMs *when* to use retrieval helps make models robust to irrelevant context and improve their overall performance, including in challenging multi-hop tasks.[1]

---

[1]Our code, data, and models are available at `https://github.com/oriyor/ret-robust`.

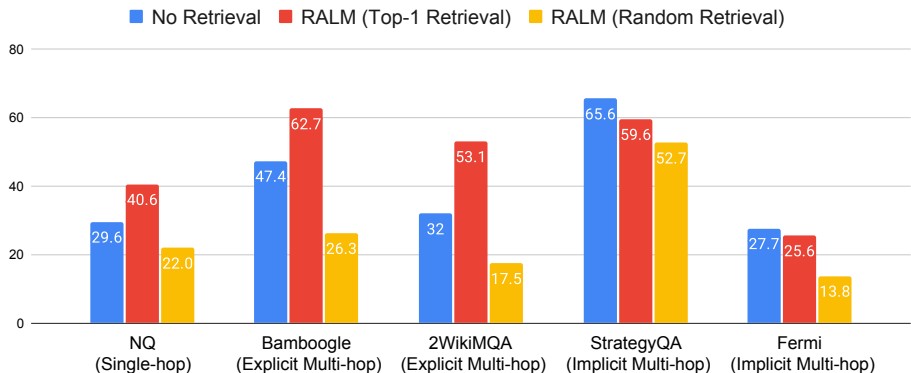

Figure 2: Accuracy for *Llama-2-13B* few-shot prompted on five QA tasks, in three settings: (a) without retrieval, (b) with top-1 retrieval from a strong search engine, and (c) with a randomly-retrieved passage. Retrieval augmentation can boost performance, but even strong retrieval hurts performance on StrategyQA and Fermi, and random contexts reduce performance dramatically.

## 2 MAKING RALMS ROBUST TO IRRELEVANT CONTEXTS

We now present our methods for building RALMs that are robust to irrelevant contexts. We begin by describing the common approach for incorporating evidence into RALMs. Next, we explore a natural baseline for using an NLI model to identify irrelevant contexts. Last, we describe our procedure for finetuning models to be robust to irrelevant context.

**In-context RALMs** Language models define a probability distribution over sequences of tokens, with *auto-regressive models* assigning a probability via next-token prediction: $p_{LM} = \Pi_{i=1}^{n} p_{\theta}(x_i | x_{<i})$, where $x_{<i}$ is the sequence of tokens preceding $x_i$ at each step and $\theta$ denotes the parameters of the LM. For RALMs, we follow the definition of *in-context RALMs* from Ram et al. (2023), where context sentences are retrieved from a corpus $C$, and generation is conditioned on the retrieved context. Given the retrieval operation $R_C$, this can be formalized as $p_{\text{RALM}} = \Pi_{i=1}^{n} p_{\theta}(x_i | R_C(x_{<i}); x_{<i})$, where $[R_C(x_{<i}); x_{<i}]$ denotes the concatenation of the retrieved evidence with the generated sequence. Generation in LMs and RALMs can also be conditioned on additional input, which we omit for brevity.

In our setting, we focus on RALMs for ODQA. We follow recent approaches such as Self-Ask and IR-CoT (Press et al., 2023; Trivedi et al., 2023; Yoran et al., 2023), for interleaving retrieval with multi-hop question answering (see Fig. 3). Retrieval is performed for every intermediate question and each context is prepended to the question. In the single-hop setting, the model has to generate the answer given a question and retrieved context. In the multi-hop setting, the model has to generate intermediate questions and answers until arriving at the final answer and the retriever is called for the original question and after each intermediate question. Formally, $x$ in this case is the generated decomposition until an intermediate step and $R_C(x)$ are the retrieved contexts for all questions in $x$.

### 2.1 IDENTIFYING IRRELEVANT CONTEXTS WITH NLI MODELS.

NLI models (Dagan et al., 2006; Bowman et al., 2015) classify whether a textual *hypothesis* is entailed, neutral, or contradicted given a textual *premise*. Recent work successfully used NLI models to automatically identify hallucinations (Honovich et al., 2022) and statement attribution (Bohnet et al., 2023) when presented with a context and generated text. Similarly, a natural baseline is to frame irrelevant context identification as an NLI problem, by using the retrieved context only when the hypothesis (i.e., final answer and intermediate question-answer pairs; Fig. 3) are classified as entailed by the premise (i.e., the retrieved context). We use a simple *back-off* strategy where we generate twice, once with $p_{LM}$ and once with $p_{RALM}$, and only use the RALM if the NLI model classified all generated answers (and intermediate questions) as entailed by the retrieved evidence.

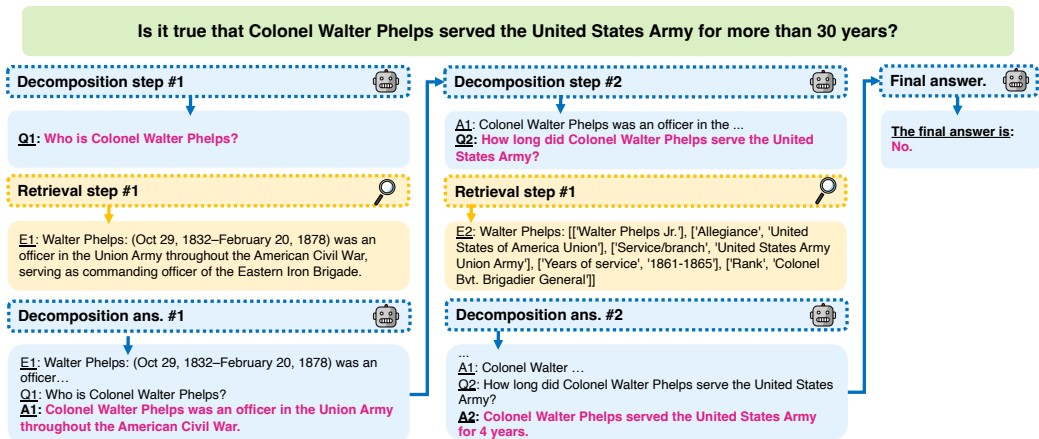

Figure 3: Interleaving decomposition and retrieval in Self-Ask format (Press et al., 2023). The model generates intermediate questions and answers until generating the final answer (model generations are shown in pink). Retrieved evidence for intermediate questions is prepended at each step.

For example, in Fig. 1, the retrieved evidence *"Jason Gerhardt... is an American actor... known for playing Cooper Barrett..."* serves as the *premise* while the question and generated answer, *"Q: Who is the actor playing Jason on general hospital? A: Steve Burton"* are concatenated and serve as our *hypothesis*. As this context is irrelevant, we expect the NLI model to label the hypothesis as *contradicting*. Given a contradicting or neutral hypothesis, we will use the standard LLM without the (potentially distracting) retrieved context. For multi-hop questions (as in Fig. 3), we additionally verify that *each* intermediate-answer pair is entailed by the retrieved evidence using all retrieved evidence as our premise and the intermediate question-answer pair as the hypothesis. For example, *"Q: Who is Colonel Walter Phelps? A: Colonel Walter Phelps was an officer in the Union Army throughout the American Civil War."* for the first intermediate question in Fig. 3.

## 2.2 TRAINING ROBUST RALMS

As in-context RALMs are not trained to use retrieved passages, a more effective solution than post-hoc filtering (using NLI) may be to train RALMs to ignore irrelevant contexts. We are interested in testing whether training on a relatively small dataset (several hundreds of examples) would suffice.

**Automatically Generating Training Data**   Our goal is to teach RALMs to be robust to irrelevant context in an ODQA setting. In the single-hop setting, generating training data is straightforward. Given access to a dataset of question-answer pairs $\{(q, a)\}$ (i.e., without contexts) and a retriever $R_C$, we use the retriever to augment questions with retrieved context. To create training examples with *relevant* contexts, we return the top-1 context from $R_C$, and for *irrelevant* contexts, we either return a low-ranked result from $R_C(q)$ or a random context (i.e., $R_C(q')$ for another question $q'$). We denote the chosen context by $r_q$. Then, the training dataset is defined by $D = \{([r_q; q], a)\}$.

Our main challenge is generating training examples for multi-hop questions. In these questions the model generates a decomposition, consisting of intermediate questions and answers, before arriving at the final answer, while the retriever is called multiple times (Fig. 3). Our goal is to automatically generate retrieval-augmented decomposition steps, $D = \{([r_x; x], y)\}$, where: $y$ is the correct generation for each step (i.e., the correct intermediate question, intermediate answer, or final answer); $x$ consists of the previously generated steps up to $y$; $r_x$ is the retrieved contexts for all steps in $x$. Our first step to automatically generate decompositions is to prompt a strong LLM without access to retrieval and to verify its answers. However, the LLM may arrive at the correct answer using an incorrect decomposition, for example in binary or comparison questions. Hence, we need to ensure the quality of generated decompositions. For multi-hop datasets which provide intermediate answers, we simply filter out generated decompositions that do not contain them. When intermediate answer annotations are unavailable, we sample from the LLM that generated the decomposition multiple times and verify self-consistency (Wang et al., 2023). Further details are given in §3.2.3.

| Dataset | Type | Example |
|---------|------|---------|
| NQ | Single-hop | What episode of law and order svu is mike tyson in? |
| 2WIKIMQA | Explicit | Where was the place of death of Isabella Of Bourbon's father? |
| BAMBOOGLE | Explicit | What is the maximum airspeed (in km/h) of the third fastest bird? |
| STRATEGYQA | Implicit | Can Arnold Schwarzenegger deadlift an adult Black rhinoceros? |
| FERMI | Implicit | How many high fives has Lebron James given/received? |

Table 1: The QA datasets in our experiments.

**Training** We use our automatically generated data $D$ to fine-tune models for generating $y$ conditioned on $[r_x; x]$ with standard maximum likelihood. Since we are mostly interested in the low-data regime, we limit the number of questions in $D$ to 1,000 in the single-hop setting and 500 in the multi-hop setting (splitting multi-hop questions to multiple examples for each step), and use parameter efficient fine-tuning (Dettmers et al., 2023). Thus, training all our models takes no more than a few hours. Additional experimental details are in §3 and §A.1.

## 3 EXPERIMENTAL SETTING

### 3.1 DATASETS

We experiment with both single- and multi-hop QA datasets. We list and give an example from each dataset in Tab. 1. Our QA benchmarks can be categorized based on their required reasoning skills:

- **Single-hop:** Information-seeking questions that do not require decomposition. We use the popular Natural Questions (NQ) dataset (Kwiatkowski et al., 2019).
- **Explicit Reasoning:** Multi-hop questions where reasoning is explicitly expressed in the question. We include 2WIKIMQA (Welbl et al., 2018) and BAMBOOGLE (Press et al., 2023).
- **Implicit Reasoning:** Mutli-hop questions where generating reasoning steps requires common-sense (implicit reasoning, Geva et al. (2021)). Such questions may have multiple valid reasoning chains. We evaluate on STRATEGYQA (Geva et al., 2021) and FERMI (Kalyan et al., 2021).

For evaluation, we follow prior work and use EM for NQ and STRATEGYQA, and $F_1$ for 2WIKIMQA and BAMBOOGLE. For FERMI, we use the official order-of-magnitude evaluation ( Kalyan et al. 2021). Following prior work (Khattab et al., 2022; Trivedi et al., 2023; Yoran et al., 2023), we evaluate on 500 random examples from the development set of each dataset. We provide additional technical details on evaluation in §A.2.

### 3.2 MODELS

We next describe our retrievers (§3.2.1), prompted baselines (§3.2.2), and finetuned models (§3.2.3).

#### 3.2.1 RETRIEVERS

Our models use a retriever based on GOOGLE SEARCH,[2] as well as the open-source COLBERTV2 (Khattab & Zaharia, 2020). Since the corpus for our datasets is Wikipedia, we format search queries as "`en.wikipedia.org` $q_i$" when accessing GOOGLE SEARCH. For COLBERTV2 our corpus is the 2018 Wikipedia from Karpukhin et al. (2020). To simulate different types of noise, we return either the top-1, a low-ranked relevant evidence,[3] or a random passage that is the top-1 evidence for a different question or intermediate question from the same dataset.

---

[2] We query Google search via the SerpAPI service: `https://serpapi.com/`.

[3] For GOOGLE SEARCH, we use the lowest returned result from the API, which is at rank 9.3 on average. For COLBERTV2 we only experiment with top-1 results.

### 3.2.2 FEW-SHOT PROMPTED BASELINES

Our main baselines are *Llama-2-13B* models prompted for QA in the Self-Ask format through in-context learning (Brown et al., 2020) with 4-6 exemplars. We also evaluate with *Llama-2-70B* on NQ. Our baselines differ based on the retrieved contexts in the exemplars (Full prompts in §A.5):

- **Self-Ask No Retrieval (SA-NR):** Exemplars are gold decompositions *without* retrieved evidence. We use this prompt to evaluate the performance of models without retrieval, when relying solely on their parametric memory, i.e, the information encoded in the model's parameters. As an additional baseline, we use this non-retrieval prompt, but still apply retrieval during inference.
- **Self-Ask Retrieval@1 (SA-R@1):** Exemplars are gold decompositions prepended with the most relevant evidence retrieved from GOOGLE SEARCH for each step.
- **Self-Ask Retrieval@10 (SA-R@10):** Exemplars are gold decompositions prepended with the lowest rank passage from Google (which is rank 10 in most cases).
- **Self-Ask Random Retrieval (SA-RMix)** Exemplars are gold decompositions prepended with either the top-1 or lowest-ranked evidence from GOOGLE SEARCH, interchangeably.

**NLI-based Models**   We use a BART-Large model (Lewis et al., 2020a) with 407 million parameters trained on the MNLI dataset (Williams et al., 2018).[4] We consider a question-answer pair as entailed if the probability for the entailment label is $\geq 0.5$. All few-shot prompted baselines have a variant with NLI, termed, SA-\*-NLI. When there is no entailment, we use the generation from the SA-NR model, which uses only the parametric memory as the back-off strategy.

### 3.2.3 FINE-TUNED MODELS

We finetune *Llama-2-13B* on 3 ODQA benchmarks, one single-hop (NQ, 1000 training examples), one explicit (2WIKIMQA, 500 questions, 1,539 examples), and one implicit (STRATEGYQA, 414 questions, 1,584 examples). Training hyperparameters are in §A.1.

**Data Generation**   We use a LLM to verify questions are answerable and to generate decompositions.[5] This is done with GPT-3, *code-davinci-002* (Brown et al., 2020; Chen et al., 2021) with 175B parameters. We prompt the model to generate decompositions using the SA-NR prompt. 2WIKIMQA contains intermediate answers, and we use those to verify generated decompositions. For the implicit STRATEGYQA we utilize only the final answer, and thus use self-consistency, as explained in §2. We sample 5 decompositions per question (one with greedy decoding and four with temperature $0.7$) and only keep the greedily-decoded decomposition when all decompositions lead to the same correct answer. To verify the quality of the generated decompositions, we manually examine 50 decompositions per dataset and find that the generated decompositions are correct in about $90\%$ of the time for STRATEGYQA and more than $95\%$ for 2WIKIMQA. As FERMI and BAMBOOGLE contain less than 300 examples, we use them exclusively for evaluation and do not include them in these experiments.

**Incorporating Retrieved Evidence in Training Examples**   To make sure the model is exposed to relevant and irrelevant context, we use either the top-1, low-ranked, or random evidence with equal probability at each step. We term the trained model SA-RetRobust. We include ablations where training is without retrieved context (SA-NoRet) or only with the top-1 evidence (SA-Ret@1).

## 4  RESULTS

Fig. 4 presents our main results, evaluating the effect that retrieving top-1 result from GOOGLE SEARCH has on the following RALMs: (a) an In-Context RALM, prompted with the SA-RMix prompt (leftmost yellow), (b) the same model, but using NLI models to identify irrelevant context (center, green), and (c) our proposed SA-RetRobust, a RALM fine-tuned on a mixture of relevant

---

[4]We use the model from `https://huggingface.co/facebook/bart-large-mnli`.

[5]To not train our models to hallucinate, we also filter single-hop questions where *code-davinci-002* fails to generate the correct answer. However, we do not fully guarantee that the gold answer appears in the retrieved context or encoded in parameters of the model being trained.

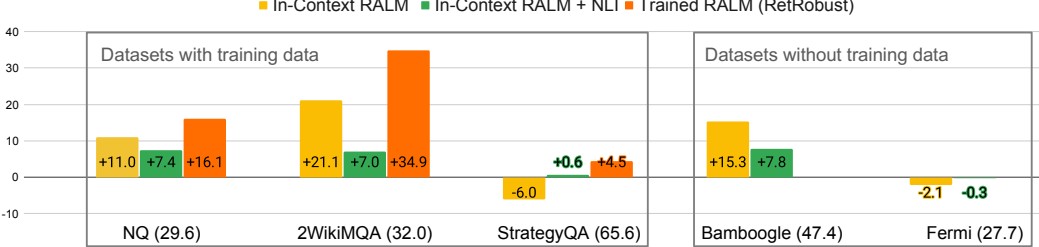

Figure 4: Results for our models on all evaluation datasets when retrieving top-1 results from GOOGLE SEARCH. Bars show the difference in performance from a model with no retrieval (whose performance is given in parenthesis for each dataset). Prompting models to use retrieval in-context (leftmost bar) increases performance on single-hop and explicit datasets, but decreases performance on implicit ones (STRATEGYQA and FERMI). When using NLI models to identify irrelevant evidence (center bar), retrieval never hurts, at a cost to gains received when retrieval is helpful. Our trained RALMs (rightmost column) outperform all other models when applicable for NQ, 2WIKIMQA, and STRATEGYQA (see §3.2.3 for more details on data generation).

and irrelevant contexts (rightmost, orange). The bars show the difference in performance from our few-shot prompted model without retrieval (whose performance is shown in parenthesis for each dataset). For the In-Context RALM, we observe that retrieval helps on NQ, 2WIKIMQA and BAMBOOGLE but reduces performance on the implicit STRATEGYQA and FERMI. Adding NLI to identify irrelevant context ensures that retrieval does not hurt, but gains are limited. Training with retrieval leads to gains across the board. We observe similar trends with the COLBERTV2 retriever, albeit at an overall decrease in accuracy (§A.3, Tab. 3.)

**Exploring the Robustness of Models to Irrelevant Context** Fig. 5 present results when simulating retrieval of irrelevant/noisy context, either by retrieving low-ranked passages (top) or random ones (bottom). When retrieving random passages, the performance of the In-Context RALM drops by more than 10 points on average, a phenomenon that can be mitigated by using NLI models. SA-RetRobust performs best across all settings. To verify that these improvements indeed stem from robustness to irrelevant context rather than task-specific training, we compare SA-RetRobust to an ablated variant trained and evaluated without retrieval (full results in Tab. 4, §A.3). SA-RetRobust is able to perform similarly to this model (within one standard deviation) when retrieving random contexts. Interestingly, when retrieving low-ranked results, SA-RetRobust outperforms the ablated model by 3.8 and 2.8 points on NQ and 2WIKIMQA, while performing only slightly worse (within a 1.2 point difference) on STRATEGYQA. Overall, our results suggest SA-RetRobust learned to both better utilize retrieval and ignore irrelevant context.

**Adding Retrieval to In-context Exemplars can Hurt Performance** Tab. 2 and Tab. 3 in §A.3 present full results with the GOOGLE SEARCH and COLBERTV2 retrievers. Interestingly, providing exemplars with retrieval performs worse than providing exemplars without retrieval, i.e., the SA-NR prompt leads to better performance even when retrieval is performed at inference time. This SA-NR prompt consistently outperforms the prompts with retrieval (SA-R@1, SA-R@10, and SA-RMix) when retrieving the top-1 result from COLBERTV2 or random contexts from GOOGLE SEARCH. In addition, the SA-R@1 model that contains top-1 results in the prompt is not the best performing even when retrieving top-1 results at inference time, losing to SA-NR by more than 2 points on average across datasets. When retrieving noisy contexts at inference time, SA-R@1 is outperformed by the other models, suggesting that showing examples for retrieval during in-context learning has a negative effect that causes *over-utilization of irrelevant context*. We observe a similar trend with *Llama-2-70B* in §A.3, Tab. 6.

**Effect of NLI** When retrieving random contexts or evaluating on the implicit STRATEGYQA and FERMI, NLI variants consistently perform best, suggesting small NLI models are sufficient to identify irrelevant evidence (Tab. 2 and Tab. 3 in §A.3). However, they reduce performance in cases

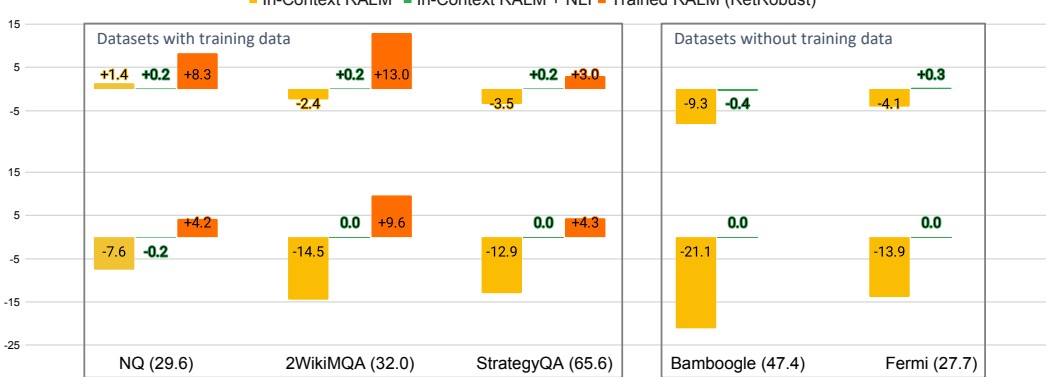

Figure 5: Results with low-rank (top) and random retrieval (bottom). Models are similar to those in Fig.4. Performance significantly decreases for the prompted model in all settings, while it is maintained when using NLI models. Our finetuned SA-RetRobust is best performing in all settings. We show that SA-RetRobust learned to both ignore irrelevant context and better utilize relevant context by comparing to an ablated model without retrieval in §4.

retrieval is helpful, e.g., on the explicit 2WIKIMQA and BAMBOOGLE. We perform a detailed analysis for our NLI variants in §5.

**Results with Finetuned Models**  Fig. 4 and Fig. 5 show SA-RetRobust consistently outperforms other models. In §A.3, Tab. 4, we present all results for all trained models, showing SA-RetRobust outperforms our ablated baselines. Specifically, it outperforms SA-NoRet (fine-tuned without retrieval) by 2.7, 2.4, and 2.4 points on average when using the top-1, a low-ranked, or a random context from GOOGLE SEARCH during inference, and SA-@1 by 0.2, 0.4, 3.2 points respectively. When retrieving top-1 results from COLBERTV2, SA-RetRobust outperforms SA-NoRet and SA-@1 by 2.7 and 0.3 points on average, respectively. Our results suggest that training on a mixture of relevant and irrelevant contexts is necessary for robustness and improved performance. We provide a study on the generalization of our trained models to other settings in §A.3.

**Results with *Llama-2-70B***  We compare SA-RetRobust with *Llama-2-70B* on the NQ dataset to assess whether larger models are more robust to irrelevant contexts. Without retrieval, the prompted *Llama-2-70B* outperforms the trained *Llama-2-13B* by 4.3 points (38.4 vs 34.1). However, when retrieving the top-1 results from GOOGLE SEARCH, SA-RetRobust outperforms all prompted *Llama-2-70B* variants by at least 3.3 points (45.7 vs 42.4), suggesting that increasing model size alone is not sufficient to make models better utilize retrieval. We provide the full results in §A.3, Tab. 6.

## 5 ANALYSIS

**When Does Irrelevant Context Cause Errors?**  To assess errors caused by irrelevant context, we manually looked at examples from NQ, 2WIKIMQA and STRATEGYQA, where models succeed without retrieval, but fail with it. Specifically, we look at examples where the model is prompted with the SA-RMix prompt that includes both top-1 and low-ranked retrieved result and is presented with low-rank or random retrieved evidence during inference. We manually annotated 40 examples in each setting (240 overall), and find that automatic errors indeed correlate with cases in which retrieval augmentation caused the model to err in 73% of the cases (65%-85% in each setting). We provide additional details and statistical tests in §A.4.

We then take a deeper look at the errors. For NQ we find that when using low-ranked context, the wrong generated answer entity appears in the retrieved context in the majority (77%) of the cases, but only in 37% when retrieving random contexts. This suggests that irrelevant context can cause errors even when the generated entities are not retrieved, as shown in §A.4, Fig. 6. For multi-hop questions, we test whether irrelevant context leads to errors in question decomposition, or in answering intermediate questions. We find that when retrieving low-ranked passages, most of the errors

(68%) for the explicit 2WIKIMQA are in intermediate *answers*, contrary to the implicit STRAT-EGYQA were errors are more prevalent in intermediate *questions* (77% of the cases, we provide an example in §A.4, Fig. 7). Similarly, when retrieving random contexts, most errors (60%) for 2WIKIMQA are in intermediate questions. This suggests that irrelevant context can cause errors in generating both an answering strategy and the answer itself, depending on the task and the retrieved context.

**When Do NLI Models Fail?**   As shown in §4, NLI models are efficient at identifying relevant context, at a cost to gains when retrieval is helpful. To better characterize NLI models, we look at the accuracy for our SA-*-NLI models as a function of the probability that the NLI model assigns to the entailment label. Tab. 8 in §A.4 shows that there are many cases where the probability for entailment is low but retrieval helps for NQ and 2WIKIMQA.

To better identify the source for such errors, we manually analysed 25 examples for each dataset where entailment was low, but retrieval augmentation led the SA-RMix model to generate the correct answer.[6] In about half of the cases the NLI model erred and the generated text is indeed entailed from the retrieved contexts. In the remaining examples, for at least a third of the cases the generated answer or decomposition is correct, but the retrieved context does not directly entail the generation. This can be partially explained by the ability of models to combine retrieval and their parametric knowledge (Talmor et al., 2020; Zhong et al., 2023; Cohen et al., 2023). We are hopeful that these results can inspire future work to focus on additional aspects of retrieval augmentation, such as the effect augmentation has on generation probability rather than on direct entailment.

## 6  RELATED WORK

Recent work has shown that the performance of LLMs can be affected by irrelevant context. Amongst others, Jia & Liang (2017); Petroni et al. (2020); Creswell et al. (2023) show that adding random or irrelevant context can decrease QA performance. This has been shown in many settings, including but not limited to factual reasoning (Kassner & Schütze, 2020; Pandia & Ettinger, 2021; Misra et al., 2023), text generation about new entities (Onoe et al., 2022), or even code generation (Jones & Steinhardt, 2022). In the context of arithmetic reasoning, Shi et al. (2023) showed that adding irrelevant context to exemplars or task specific instructions can help, suggesting the model may be equipped with such skills from pre-training. Other methods try to reduce the number of retrieval calls, by focusing on cases where confidence is low (Jiang et al., 2023) or retrieving information for rare entities (Mallen et al., 2023). Closest to our work is that of Li et al. (2023) that propose LLMs with a "controllable memory" that will enable them to ignore irrelevant context. However, their LLMs are finetuned on over 200K training examples, where our focus is on performance when training with 1,000 questions or less, and training data is automatically generated. In addition, we focus on a multi-hop QA setting, where the retriever is called multiple times (§2).

A similar line of work focuses on when models should use parametric or retrieved knowledge, especially when there are conflicts (Longpre et al., 2021; Chen et al., 2022). It has been recently proposed to train models to generate from both parametric and retrieved knowledge (Neeman et al., 2023) or make better use of in-context exemplars (Zhou et al., 2023).

## 7  CONCLUSION

In this work, we provide a thorough analysis showing current RALMs are not robust to irrelevant retrieved context, causing them to perform worse on certain tasks. In cases where training is not possible, a simple NLI baseline is efficient to increase robustness, at a cost of discarding relevant passages. When training is possible, we introduce an automatic data generation framework for single and challenging multi-hop tasks, and show training on as few as 1,000 examples with intentionally varied quality suffice to make models robust to irrelevant context and improve overall performance. While our focus in this work is on in-domain settings, we are hopeful our work could inspire future research towards a general RALM that is robust to irrelevant context.

---

[6]There are only 25 such examples for the NQ dataset.

ACKNOWLEDGEMENTS

We would like to our colleagues at TAU NLP for their insightful comments. We thank SerpAPI for their support by granting us an academic discount. This research was partially supported by the Yandex Initiative for Machine Learning and the European Research Council (ERC) under the European Union Horizons 2020 research and innovation programme (grant ERC DELPHI 802800). This work was completed in partial fulfillment of the Ph.D. of Ori Yoran.

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

## A APPENDIX

### A.1 MODELS

**Llama-2** In all cases, we use the vanilla variant of the *Llama-2* models from `https://huggingface.co/meta-llama`, with half precision.

**Decomposition Generation** Questions in our multi-hop datasets require between 2-4 decomposition steps. Hence we limit the number of generation steps to 5. In Tab. 8 we show that the number of cases in which the model does not arrive at an answer in 5 steps, termed as failures, is very small when generating with top-1 results from GOOGLE SEARCH, at 0.4% for 2WIKIMQA and 1.2% for STRATEGYQA. Failures are much higher when retrieving random contexts, at 37.0% for 2WIKIMQA and 34.4% for STRATEGYQA. These are usually cases the model enters an infinite loop. Following recent work, (Wang et al., 2023; Yoran et al., 2023) we use greedy decoding when generating decompositions.

**Training** We fine-tune all our models with QLoRA (Dettmers et al., 2023) for parameter efficient fine-tuning. We use the default hyperparameters from `https://github.com/daniel-furman/sft-demos/blob/main/src/sft/one_gpu/llama-2/guanaco/sft-llama-2-13b-guanaco-peft.ipynb`. We train all our models for 5 epochs, with a learning rate of $2e - 4$ and linear scheduling on a single GPU. The training time for each model was no longer than 3.5 hours.

## A.2 EVALUATION

In some cases, the models do not arrive at a final answer (§A.1). In such cases, we assign a score of $0.5$ for STRATEGYQA and $0$ for all other datasets. For FERMI, following past work (Yoran et al., 2023), we use all 286 "Real Fermi Problems" for evaluation and provide the gold answers measure units (meters, cubes, litres, etc...) as additional input to our models .

## A.3 FULL RESULTS

Tab. 2 and Tab. 3 presents the full results for our prompted models with GOOGLE SEARCH and COLBERTV2, respectively. Tab. 4 presents full results for all our trained models, averaged over three seeds. Tab. 6 presents results for *Llama-2-70B* on NQ with the GOOGLE SEARCH retriever.

**Out of Distribution Generalization** To test the generalization of our trained models in an out of distribution (OOD) setting, we trained a version of our models on a mixture of our STRATEGYQA and 2WIKIMQA data and evaluate on BAMBOOGLE and FERMI. Since the evaluation task can differ from the training data (for example in FERMI the model needs to generate an equation before the final answer), we provided the models with one exemplar during inference. We provide the full results for this experiment in Tab. 5. We note that the standard deviation in these experiments is larger than in Table 3, probably due to the small support size at 120 for BAMBOOGLE and 286 for FERMI. Still, when comparing between the trained models, SA-RetRobust is either the best performing model or within one standard deviation in all settings. However, we also observe some surprising trends that may be related to a failure of the model to generalize or to the effect of the in-context exemplar: (a) For BAMBOOGLE, when not using a retriever, the model prompted and evaluated without retrieval outperforms the model trained without retrieval (47.4 vs 40.8), and (b) For FERMI, we see a slight decrease in accuracy from the model trained and evaluated without retrieval to our trained SA-RetRobust model when evaluating with low-ranked or random retrieval (29.3 vs 27.9 and 27.6 respectively). Overall, we are hopeful that these results will help future research towards a general RALM that is robust to irrelevant context.

## A.4 ANALYSIS

For our study regarding cases irrelevant context caused SA-RMix to err, we annotate examples with the following categories (a) *Valid*: the prediction is a paraphrase of the correct answer or a plausible answer to an ambiguous question (b) *Wrong*: the prediction with retrieval is wrong and the prediction without retrieval is correct, (c) *Both Wrong*: the prediction with retrieval is wrong, but the prediction without retrieval was also wrong (due to bad decomposition that can spuriously lead to a correct answer in binary or comparison questions). We provide the full result in Tab. 7. We verify our results are statistical significant by running a binomial test for the hypothesis: "Most cases where automatic metrics decrease by the introduction of irrelevant context are not actual errors" which was rejected with p-value<0.01.

Fig. 6 presents an example where irrelevant context causes *Llama-2-13B* to err when the generated entity does not appear in the retrieved context. Fig. 7 shows an example where random retrieval caused the model to generate a bad strategy in STRATEGYQA and Tab. 8 presents the full results for our analysis of NLI models.

## A.5 PROMPTS

We provide our SA-NR, SA-R@1, and SA-R@10 prompts for NQ in Tab. 8, Tab. 9, Tab. 10, respectively. For the SA-RMix prompt, we use exemplars form the SA-R@1 and SA-R@10 prompts,

| Dataset | Inference Retrieval | NR | NR -NLI | R@1 | R@1 -NLI | R@10 | R@10 -NLI | RMix | RMix -NLI |
|---|---|---|---|---|---|---|---|---|---|
| NQ | None | 29.6 | n/a | n/a | n/a | n/a | n/a | n/a | n/a |
| | @1 | **41.0** | 38.4 | 39.0 | 36.4 | **41.0** | 36.8 | 40.6 | 37.0 |
| | @10 | 30.2 | 29.8 | 25.6 | 29.4 | 30.0 | **31.0** | **31.0** | 29.8 |
| | Random | 28.2 | **29.6** | 17.2 | 29.4 | 22.2 | 29.4 | 22.0 | 29.4 |
| 2WikiMQA | None | 32.0 | n/a | n/a | n/a | n/a | n/a | n/a | n/a |
| | @1 | **56.0** | 39.9 | 51.6 | 38.3 | 51.6 | 39.2 | 53.1 | 39.0 |
| | @10 | **33.0** | 32.2 | 27.5 | 32.5 | 30.9 | 32.3 | 29.6 | 32.2 |
| | Random | 27.0 | 32.0 | 13.7 | 32.0 | 21.3 | **32.2** | 17.5 | 32.0 |
| StrategyQA | None | 65.6 | n/a | n/a | n/a | n/a | n/a | n/a | n/a |
| | @1 | 62.1 | 65.6 | 63.8 | **66.7** | 61.4 | 65.8 | 59.6 | 66.2 |
| | @10 | 60.4 | 65.6 | 61.0 | 65.6 | 60.5 | 65.4 | 62.1 | **65.8** |
| | Random | 58.4 | **65.6** | 53.4 | **65.6** | 57.0 | **65.6** | 52.7 | **65.6** |
| Bamboogle | None | 47.4 | n/a | n/a | n/a | n/a | n/a | n/a | n/a |
| | @1 | 68.0 | 55.9 | 61.2 | 56.0 | **68.9** | 58.0 | 62.7 | 55.2 |
| | @10 | 41.4 | **47.4** | 32.1 | 45.9 | 44.5 | 45.9 | 38.1 | 47.0 |
| | Random | 39.5 | **47.4** | 24.7 | **47.4** | 34.8 | **47.4** | 26.3 | **47.4** |
| Fermi | None | 27.7 | n/a | n/a | n/a | n/a | n/a | n/a | n/a |
| | @1 | 27.4 | **28.2** | 25.2 | 27.6 | 27.5 | 27.7 | 25.6 | 27.4 |
| | @10 | 24.0 | 27.7 | 27.1 | 27.6 | 25.1 | 27.7 | 23.6 | **28.0** |
| | Random | 22.1 | **27.7** | 17.2 | **27.7** | 17.4 | **27.7** | 13.8 | **27.7** |

Table 2: Full results for our prompted *Llama-2-13B* models with the GOOGLE SEARCH retriever.

| Dataset | Inference Retrieval | NR | NR -NLI | R@1 | R@1 -NLI | R@10 | R@10 -NLI | RMix | RMix -NLI |
|---|---|---|---|---|---|---|---|---|---|
| NQ | None | 29.6 | n/a | n/a | n/a | n/a | n/a | n/a | n/a |
| | @1 | 34.6 | **34.8** | 31.2 | 33.2 | 32.4 | 33.8 | 32.8 | 33.8 |
| 2WikiMQA | None | 32.0 | n/a | n/a | n/a | n/a | n/a | n/a | n/a |
| | @1 | **42.2** | 36.2 | 37.3 | 34.9 | 36.7 | 35.0 | 39.6 | 35.3 |
| StrategyQA | None | 65.6 | n/a | n/a | n/a | n/a | n/a | n/a | n/a |
| | @1 | 61.6 | **66.0** | 64.3 | 65.1 | 61.1 | 64.9 | 61.6 | 64.7 |
| Bamboogle | None | 47.4 | n/a | n/a | n/a | n/a | n/a | n/a | n/a |
| | @1 | **50.0** | 48.6 | 37.4 | 46.6 | 38.1 | 47.4 | 38.2 | 48.7 |
| Fermi | None | 27.7 | n/a | n/a | n/a | n/a | n/a | n/a | n/a |
| | @1 | 25.9 | 27.3 | 23.2 | 27.8 | 21.2 | **28.0** | 24.4 | **28.0** |

Table 3: Full results for our prompted *Llama-2-13B* models with the COLBERTV2 retriever.

interchangeably. We add a small instruction for the QA task before the exemplars. Our prompts contain 6 exemplars for NQ, 2WIKIMQA, and STRATEGYQA, 5 for FERMI, and 4 for BAMBOOGLE. All our prompts are publicly available, together with our models, data, and code.

| Dataset | Retriever | Inference | SA-NoRet | SA-Ret@1 | SA-RetRobust |
|---|---|---|---|---|---|
| NQ | None | None | 34.1±0.8 | n/a | n/a |
| | Google | @1 | 42.8±0.8 | **46.3±0.6** | 45.7±0.6 |
| | Google | @10 | 37.0±1.0 | **38.2±0.6** | 37.9±0.5 |
| | Google | @Random | 31.1±0.1 | 31.4±0.5 | **33.8±0.2** |
| | ColBERTV2 | @1 | 41.5±0.4 | **43.5±0.2** | **43.5±0.6** |
| 2WikiMQA | None | None | 42.2±0.6 | n/a | n/a |
| | Google | @1 | 64.6±0.7 | 66.7±1.0 | **66.9±1.0** |
| | Google | @10 | 40.8±0.5 | 43.9±0.3 | **45.0±0.4** |
| | Google | @Random | 40.4±0.8 | 37.5±1.0 | **41.6±0.2** |
| | ColBERTV2 | @1 | 54.4±0.7 | 57.0±0.5 | **57.6±0.5** |
| StrategyQA | None | None | 69.8±0.9 | n/a | n/a |
| | Google | @1 | 67.1±0.4 | 69.0±1.2 | **70.1±1.1** |
| | Google | @10 | 66.6±1.1 | 68.1±0.3 | **68.6±0.5** |
| | Google | @Random | 66.6±0.7 | 66.9±1.2 | **69.9±1.8** |
| | ColBERTV2 | @1 | 65.9±0.6 | 68.4±1.4 | **68.8±0.9** |

Table 4: Full results for our trained *Llama-2-13B* models. Results are averaged over three seeds. For our RALMs, we use either GOOGLE SEARCH or COLBERTV2 as our retrievers during inference.

| Dataset | Retriever | Inference | SA-NoRet | SA-Ret@1 | SA-RetRobust |
|---|---|---|---|---|---|
| BAMBOOGLE | None | None | 40.8±2.0 | n/a | n/a |
| | Google | @1 | 57.4±2.0 | 61.3±1.4 | **64.7±1.5** |
| | Google | @10 | 33.1±1.9 | 39.2±2.0 | **42.0±2.2** |
| | Google | @Random | 29.8±1.8 | 38.4±4.8 | **43.6±1.6** |
| | ColBERTV2 | @1 | 37.1±1.5 | 48.2±0.7 | **49.6±1.8** |
| FERMI | None | None | 29.3±0.4 | n/a | n/a |
| | Google | @1 | **31.3±1.2** | 29.6±0.8 | 29.2±1.6 |
| | Google | @10 | 28.3±1.5 | **28.6±2.5** | 27.9±1.9 |
| | Google | @Random | 25.3±1.3 | **27.9±2.4** | 27.6±0.6 |
| | ColBERTV2 | @1 | 28.3±0.1 | 28.9±0.4 | **30.0±1.1** |

Table 5: Full results for our trained *Llama-2-13B* models in an out of distribution setting. In this setting, our models are trained on a mixture of STRATEGYQA and 2WIKIMQA and evaluated on BAMBOOGLE and FERMI. Results are averaged over three seeds. For our RALMs, we use either GOOGLE SEARCH or COLBERTV2 as our retrievers during inference.

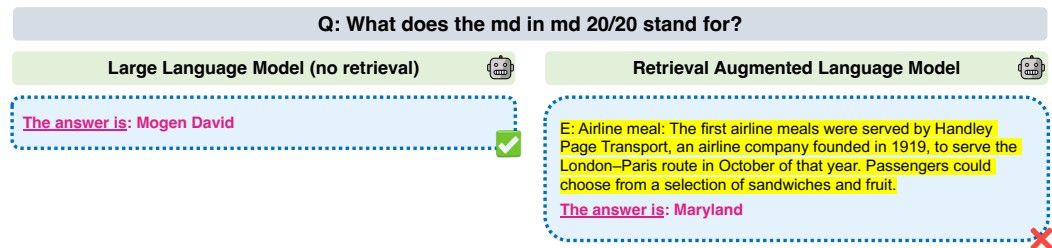

Figure 6: An example from NQ where retrieval caused *Llama-2-13B* to err, although the generated entity does not appear in the retrieved context.

| Inference Retrieval | NR | NR -NLI | R@1 | R@1 -NLI | R@10 | R@10 -NLI | RMix | RMix -NLI | SA- No-Ret | SA- RetBust |
|---|---|---|---|---|---|---|---|---|---|---|
| #Params | 70B | 70B | 70B | 70B | 70B | 70B | 70B | 70B | 13B | 13B |
| None | **38.4** | n/a | n/a | n/a | n/a | n/a | n/a | n/a | 34.1 | n/a |
| @1 | 41.4 | 41.8 | 41.2 | 42.4 | 41.6 | 42.4 | 41.2 | 42.0 | 42.8 | **45.7** |
| @10 | **38.8** | 36.2 | 30.2 | 34.2 | 33.4 | 35.4 | 31.8 | 35.2 | 37.0 | 37.9 |
| Random | 33.6 | **38.2** | 28.8 | 36.8 | 35.2 | **38.2** | 31.0 | 38.0 | 31.1 | 33.8 |

Table 6: Results for NQ with GOOGLE SEARCH and *Llama-2-70B*.

| | Inference Retrieval | Valid | Wrong | Both Wrong |
|---|---|---|---|---|
| NQ | @10 | 34% | 66% | 0% |
| | Random | 22% | 78% | 0% |
| 2WIKIMQA | @10 | 2% | 72% | 23% |
| | Random | 0% | 85% | 15% |
| STRATEGYQA | @10 | 3% | 65% | 32% |
| | Random | 0% | 70% | 30% |

Table 7: Full results for our analysis regarding cases where augmenting retrieved contexts caused *Llama-2-13B* prompted with SA-RMix to err. Classes and additional details are provided in §5.

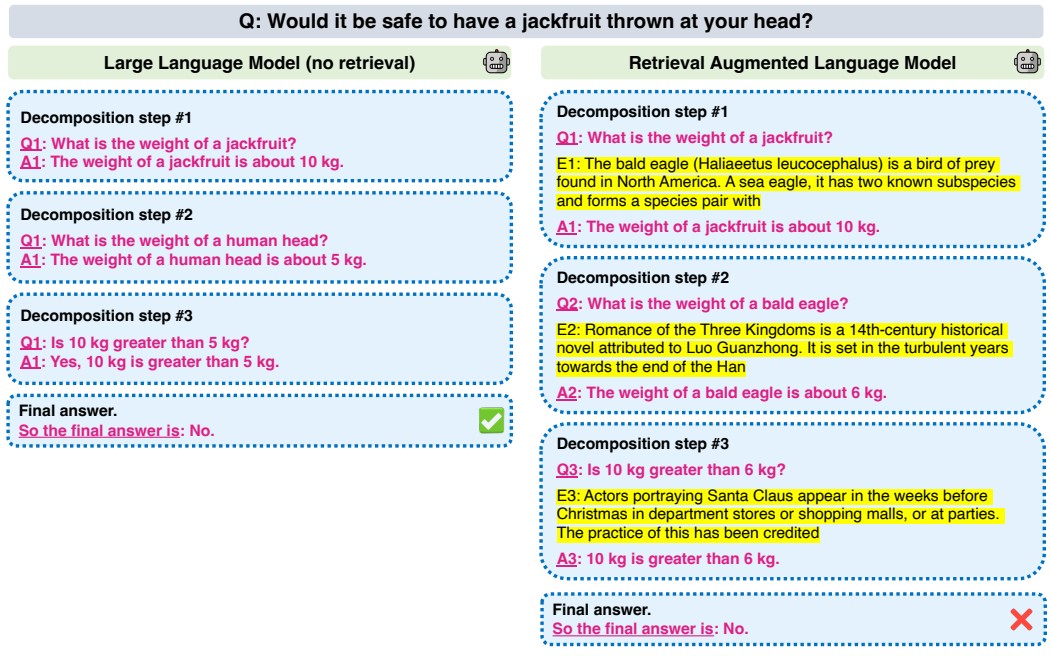

Figure 7: An example from STRATEGYQA irrelevant context causes *Llama-2-13B* to generate a wrong strategy (right). Without retrieval (left), the model succeeds in generating the correct answer.

| | Inference Retrieval | Failures % | Low-Entailment % | Δ | Med-Entailment % | Δ | High-Entailment % | Δ |
|---|---|---|---|---|---|---|---|---|
| NQ | @1 | 0.0% | 32.6% | +0.11 | 12.8% | +0.09 | 54.6% | +0.11 |
| | @10 | 0.0% | 69.4% | +0.01 | 9.4% | +0.06 | 21.2% | +0.01 |
| | Random | 0.0% | 97.2% | −0.07 | 2.2% | −0.2 | 0.6% | 0.0 |
| 2WIKIMQA | @1 | 0.4% | 83.0% | +0.12 | 5.6% | +0.34 | 11.0% | +0.55 |
| | @10 | 2.8% | 93.8% | −0.02 | 2.6% | −0.11 | 0.8% | +0.08 |
| | Random | 37.0% | 63.0% | −0.06 | 0.0% | 0.0 | 0.0% | 0.0 |
| STRATEGYQA | @1 | 1.2% | 96.2% | −0.07 | 2.4% | +0.17 | 0.2% | 0.0 |
| | @10 | 2.6% | 95.8% | −0.04 | 1.4% | 0.0 | 0.2% | 0.0 |
| | Random | 34.4% | 56.6% | −0.13 | 0.0% | 0.0 | 0.0% | 0.0 |

Table 8: Results for our NLI analysis. 'Failures' indicates that the decomposition model was not able to arrive at the answer (see §A.1). Other examples are split based on their entailment probability: low probability is $< \frac{1}{3}$, medium probability is in $[\frac{1}{3}, \frac{2}{3}]$, and high probability is $> \frac{2}{3}$. $\Delta$ indicates the improvement in accuracy when using retrieval. For NQ and 2WIKIMQA, many cases where retrieval is helpful have low entailment probability. For the implicit STRATEGYQA most examples have low entailment, but retrieval helps in the few examples with medium entailment.

---

***Given the following question, answer it by providing follow up questions and intermediate answers. If intermediate questions are not necessary, answer the question directly.***

\#
Question: how did the big red one get its name
Are follow up questions needed here: No.
So the final answer is: its shoulder patch
\#
Question: where are the cayman islands on the map
Are follow up questions needed here: No.
So the final answer is: western Caribbean Sea
\#
Question: who won the war between north korea and south korea
Are follow up questions needed here: No.
So the final answer is: technically still at war
\#
Question: when does it's always sunny in philadelphia season 13 start
Are follow up questions needed here: No.
So the final answer is: September 5, 2018
\#
Question: who sang you got a friend in me from toy story
Are follow up questions needed here: No.
So the final answer is: Randy Newman
\#
Question: when was the first person sent to space
Are follow up questions needed here: No.
So the final answer is: 12 April 1961
\#
Question:

Figure 8: The SA-NR prompt used in our NQ experiments.

*Given the following question, answer it by providing follow up questions and intermediate answers. If intermediate questions are not necessary, answer the question directly. You are provided with evidence that can help you arrive at the answer before the question.*

\#

Context1: The Big Red One: Fuller was a World War II veteran and served with the 1st Infantry Division, which is nicknamed "The Big Red One" for the red numeral "1" on the division's shoulder patch. He received the Silver Star, Bronze Star, and Purple Heart during his service.

Question: how did the big red one get its name

Are follow up questions needed here: No.

So the final answer is: its shoulder patch

\#

Context1: Location Map of Cayman Islands: The given Cayman Islands location map shows that the Cayman Islands are located in the western Caribbean Sea. Location Map of Cayman Islands. Where is Cayman ...

Question: where are the cayman islands on the map

Are follow up questions needed here: No.

So the final answer is: western Caribbean Sea

\#

Context1: Korean War — Combatants, Summary, Years, Map ... - Britannica: After more than a million combat casualties had been suffered on both sides, the fighting ended in July 1953 with Korea still divided into two hostile states. Negotiations in 1954 produced no further agreement, and the front line has been accepted ever since as the de facto boundary between North and South Korea.

Question: who won the war between north korea and south korea

Are follow up questions needed here: No.

So the final answer is: technically still at war

\#

Context1: It's Always Sunny in Philadelphia (season 13): The thirteenth season of the American comedy television series It's Always Sunny in Philadelphia premiered on FXX on September 5, 2018. ... The season consists of ...

Question: when does it's always sunny in philadelphia season 13 start

Are follow up questions needed here: No.

So the final answer is: September 5, 2018

\#

Context1: You've Got a Friend in Me: "You've Got a Friend in Me" is a song by Randy Newman. Used as the theme song for the 1995 Disney/Pixar animated film Toy Story, it has since become a major ...

Question: who sang you got a friend in me from toy story

Are follow up questions needed here: No.

So the final answer is: Randy Newman

\#

Context1: April 1961: Yuri Gagarin from the Soviet Union was the first human in space. His vehicle, Vostok 1 circled Earth at a speed of 27,400 kilometers per hour with the flight lasting 108 minutes.

Question: when was the first person sent to space Are follow up questions needed here: No.

So the final answer is: 12 April 1961

\#

Question:

Figure 9: The SA-R@1 prompt used in our NQ experiments.

*Given the following question, answer it by providing follow up questions and intermediate answers. If intermediate questions are not necessary, answer the question directly. You are provided with evidence that can help you arrive at the answer before the question.*

#
Context1: 16th Infantry Regiment (United States): As part of the new 1st Expeditionary Division, soon to become known as the 'Big Red One', the 16th Infantry, commanded by William Herbert Allaire Jr., sailed
Question: how did the big red one get its name
Are follow up questions needed here: No.
So the final answer is: its shoulder patch
#
Context1: Module:Location map/data/Cayman Islands: Module:Location map/data/Cayman Islands is a location map definition used to overlay markers and labels on an equirectangular projection map of Cayman
Question: where are the cayman islands on the map
Are follow up questions needed here: No.
So the final answer is: western Caribbean Sea
#
Context1: First Battle of Seoul: The First Battle of Seoul, known in North Korean historiography as the Liberation of Seoul, was the North Korean capture of the South Korean capital, Seoul,
Question: who won the war between north korea and south korea
Are follow up questions needed here: No.
So the final answer is: technically still at war
#
Context1: It's Always Sunny in Philadelphia (season 13): The thirteenth season of the American comedy television series It's Always Sunny in Philadelphia premiered on FXX on September 5, 2018.
Question: when does it's always sunny in philadelphia season 13 start
Are follow up questions needed here: No.
So the final answer is: September 5, 2018
#
Context1: Randy Newman – You've Got a Friend in Me Lyrics: 'You've Got A Friend In Me' is the theme song of the Toy Story franchise, recurring throughout the series in different contexts. It's first
Question: who sang you got a friend in me from toy story
Are follow up questions needed here: No.
So the final answer is: Randy Newman
#
Context1: Timeline of space exploration: This is a timeline of space exploration which includes notable achievements, first accomplishments and milestones in humanity's exploration of outer space.
Question: when was the first person sent to space Are follow up questions needed here: No.
So the final answer is: 12 April 1961
#
Question:

Figure 10: The SA-R@10 prompt used in our NQ experiments.

