# OpenReview forum: "Making Retrieval-Augmented Language Models Robust to Irrelevant Context"
_ICLR.cc/2024/Conference — ICLR 2024 poster_

### Official Review · Reviewer_Kczf · 2023-10-29

**Soundness:** 2 fair
**Presentation:** 3 good
**Contribution:** 2 fair
**Rating:** 6
**Confidence:** 4

**Summary:**

This paper presents an analysis on open-domain question answering benchmarks where retrieval harms the performance. The authors also propose two methods to mitigate such issue: 1. using an NLI model to filter out passages that do not entail question-answer pairs, and 2. automatically generating data to fine-tune the language model to be robust to irrelevant contexts. The authors show that as few as 1000 examples suffice to train the model to be robust to irrelevant context while preserving the performance.

**Strengths:**

This paper focuses on irrelevant context for open-domain question answering, which is a key and crucial part for RALMs performance. The paper proposes an automatic way to generate decomposed questions for training

**Weaknesses:**

- I am not fully convinced that the NLI model work / add any significance to the paper presentation, since there is no guarantee on the accuracy if there is no gold passage provided. With results and analysis from section 4 and 5, I don’t believe this proposed NLI model can be claimed as a significant contribution for this paper.
- section 4 can be better presented, the color coding is a bit confusing…
- For the analysis in section 5, conclusions drawn from 40 / 25 examples does not show enough statistical significance.

**Questions:**

- Additional “ranked” in section 3.2.1?
- For generating training data, it seems that the top-1 passage generated from Rc(q) is considered “relevant”, but for some harder questions, or suboptimal retriever, this is not guarantee, and even google search failed sometime for harder dataset. Although this might not be in the scope, I wonder whether there is any remedy for that?
- How do you define ambiguous question for section 5?

---

> ### Author Response · Authors · 2023-11-18
> **Response to Reviewer Kczf (1/3)**
>
> We thank the reviewer for the helpful comments. We reply in different comments due to the maximum characters limitation.
>
> >W1: I am not fully convinced that the NLI model work / add any significance to the paper presentation, since there is no guarantee on the accuracy if there is no gold passage provided. With results and analysis from section 4 and 5, I don’t believe this proposed NLI model can be claimed as a significant contribution for this paper.
>
> We thank the reviewer for the helpful comment. In our original version, we mentioned that the NLI model serves as a strong natural baseline inspired by recent work. However, this may not have been expressed thoroughly enough in the previous version. Following your comments, we made several changes to our paper to better present these models as a baseline, rather than the main proposed method. Specifically, we updated the Introduction, Section 2, and Conclusion to highlight this message.
>
> Contrary to previous work, our focus is on systematically evaluating the effect of NLI models to make models robust to irrelevant context, including in multi-hop tasks. We believe that our results that show NLI models can increase robustness at a performance cost are interesting for the community and that our analysis regarding when and how these failures happen can inspire future research.
>
> >W2: section 4 can be better presented, the color coding is a bit confusing…
>
> We thank you for this helpful comment. To improve readability, we made major updates to Section 4. Mainly:
> - We updated Figures 4 and 5 to present the delta from the model prompted without retrieval instead of the absolute results. We believe this visualization better conveys the effect augmentation has in each setting. For simplicity, we also removed the model ablated for retrieval from these figures, thereby reducing the number of models shown to three.
> - We improved the visualization of Figures 4 and 5 to visually separate between datasets with and without training data.
> - We revised the entire section in order to improve its readability.

---

> ### Author Response · Authors · 2023-11-18
> **Response to Reviewer Kczf (2/3)**
>
> > W3: For the analysis in section 5, conclusions drawn from 40 / 25 examples does not show enough statistical significance.
>
> We thank the reviewer for the helpful comment and for pointing out the small number of samples used for these analyses. We added a statistical test for each claim made in the analysis (Section 5).
>
> *When Does an Irrelevant Context Cause Errors?*
>
> - The goal of this analysis is to qualitatively analyze if adding irrelevant context indeed causes models to err. To perform this analysis, we annotated **240** examples (**40** examples in **6** settings) which required many hours of human labor. We discovered that irrelevant context indeed caused errors in 73% of the cases (174/240). Therefore, we were able to reject the null hypothesis that “Most cases where automatic metrics decrease by the introduction of irrelevant context are not actual errors” with p-value=<0.01 using a binomial test, suggesting that introduction of irrelevant context indeed causes models to err. We updated the text to make the goal of this analysis clearer and added a discussion for this test in A.4.
> - Similarly, the null hypothesis: “For NQ, when retrieving low-ranked context, the wrong generated answer entity appears in the retrieved context in 50% or less of the cases” was rejected with p-value<0.01.
> - For the null hypothesis: “For NQ, when retrieving random context, the wrong generated answer entity appears in the retrieved context in 50% or more of the cases” we received a p-value of 0.071. Subsequently, we performed a t-test for the hypothesis: “For NQ, the percentage of cases the generated entity appears in the retrieved context is similar when retrieving random and low-ranked context”, which was rejected with p-value<0.01.
> - The goal of our qualitative analysis regarding the location of errors in multi-hop questions (beginning at the last paragraph of page 8) is to highlight high-level trends regarding different errors made by in-context RALMs model when decomposing complex questions. We updated the text accordingly.
>
> *When Do NLI Models Fail?*
>
> - This is a qualitative analysis aimed to highlight high-level trends that can facilitate future research.
> - We manually analyzed 25 examples for each dataset (50 overall) since there are only 25 examples for NQ (~5% of the evaluated examples) where the following three conditions are met: (a) the NLI model gives a low probability for the entailment label, (b) the model without retrieval generated the wrong answer, and (c) the In-Context-RALM generated the correct answer.
>
> While we agree that this analysis is based on a small sample size, we believe that our results are still interesting. Specifically, we are hopeful that the following observations can inspire future work: (a) current NLI models can still fail on this entailment task, and (b) retrieval augmentation can be helpful even when there is no direct entailment. We will also be happy to run our models on additional NQ examples in order to increase the  sample size for this analysis in future versions.
>
> >Q1: Additional “ranked” in section 3.2.1?
>
> Thank you for pointing us to this error. We fixed it in the updated version.
>
> >Q2: For generating training data, it seems that the top-1 passage generated from Rc(q) is considered “relevant”, but for some harder questions, or suboptimal retriever, this is not guarantee, and even google search failed sometime for harder dataset. Although this might not be in the scope, I wonder whether there is any remedy for that?
>
> We thank you for the interesting question and we believe this can be an interesting area to explore in future work. For example, one can only filter cases where the gold answer appears in the retrieved context, use QA models to identify passages that alter the predictions of RALMs to the correct answer, or use NLI models. However, none of these methods can 100% guarantee that the retrieved contexts are indeed relevant. Alternatively, we can use human annotators.
>
> Nevertheless, our results (Table 4) also suggest that examples that include irrelevant context are helpful for robustness, suggesting that filtering such examples from the training data might actually decrease performance. Experimenting with more sophisticated methods to better control the proportion of irrelevant context can be an interesting direction for future work.

---

> ### Author Response · Authors · 2023-11-18
> **Response to Reviewer Kczf (3/3)**
>
> >Q3: How do you define ambiguous question for section 5?
>
> We define ambiguous questions as questions that have multiple plausible answers, following recent work [1]. For the analysis, we consider cases where the In-Context-RALM generated a different plausible answer as valid. We updated the phrasing in the new version in A.4 to make the definition of ambiguous questions clearer
>
> [1] AmbigQA: Answering Ambiguous Open-domain Questions, Min et al., EMNLP 2020.
>
> >Conclusion
>
> In conclusion, we thank the reviewer again for their helpful review. We addressed all of the issues raised by the reviewer in the updated version. Specifically, we made major modifications to improve the Soundness and Presentation of the work. We are hopeful that, given these changes, the reviewer would consider raising their score.

---

> > ### Comment · Reviewer_Kczf · 2023-11-21
> >
> > Thank you for updating the manuscript and the clarifications! I updated my score accordingly.
> >
> > However, although it might not be in the scope of this paper, I wonder whether the NLI models that trained solely on MNLI in detecting irrelevant context on long-form QA datasets, or complex QA datasets that have inherent ambiguities (in terms of interpretations of the question, which is different than Min et al. 2020).

---

> > > ### Author Response · Authors · 2023-11-22
> > > **Feedback on Author Response**
> > >
> > > Thank you for your response and for appreciating the modifications in the new version. In our work, we show that NLI models can be used to improve robustness to irrelevant context, at a cost to discarding relevant ones (Section 4). Previous work has also shown that NLI models can be used in QA to detect hallucinations [1] and to improve attribution [2]. We will be happy to further discuss usage of NLI models in QA in future versions, and would appreciate any references that we may have missed. We believe that improving the performance of NLI models in order to improve robustness to irrelevant context can be an interesting area for future work.
> > >
> > > [1] TRUE: Re-evaluating factual consistency evaluation., Honovich et al., ACL 2022
> > >
> > > [2] Attributed Question Answering: Evaluation and Modeling for Attributed Large Language Models, Bohnet et al., 2022

---

### Official Review · Reviewer_dWp6 · 2023-10-30

**Soundness:** 3 good
**Presentation:** 3 good
**Contribution:** 3 good
**Rating:** 6
**Confidence:** 4

**Summary:**

This paper makes a systematic study about the robustness of RALM, where it identifies the potential threat resulted from irrelevant retrieved contexts. The paper further introduces two approach to confront this problem. One is to leverage a small-scale NLI model to filter out the irrelevant context. The other one is to fine-tuned the language model with  a mixture of relevant and irrelevant contexts. The paper performs comprehensive experiments on one-hop and multi-hop QA datasets to verify the proposed methods.

**Strengths:**

s1. This paper presents a through analysis for the robustness of RALM to noisy context, which is fundamentally important to the research of LLM, question answering, and information retrieval.
s2. Despite simplicity, the two proposed methods are meaningful and empirically positive.

**Weaknesses:**

w1. The use of a filtering module to mitigate contextual noise and filtering the language model with noisy context are two established approaches found in various related works on open-domain question answering and conversational question answering. While there may be variations in specific implementations, they might not be regarded as technical breakthroughs for this problem. This paper should conduct a more comprehensive investigation of related techniques.

w2. The experimental study can be improved (please refer to my posted questions).

**Questions:**

Q1. How generalizable is the fine-tuned model? If it is applied to other QA datasets, especially with different kind of context noise, what will happen?

Q2. Which part of the experiment can support the statement "models finet-uned solely with relevant contexts are far less robust than those finetuned with a mixture of relevant and irrelevant contexts"?

---

> ### Author Response · Authors · 2023-11-18
> **Response to Reviewer dWp6**
>
> >W1: The use of a filtering module to mitigate contextual noise and filtering the language model with noisy context are two established approaches found in various related works on open-domain question answering and conversational question answering. While there may be variations in specific implementations, they might not be regarded as technical breakthroughs for this problem. This paper should conduct a more comprehensive investigation of related techniques.
>
> Thank you for your comment. While we mentioned that our filtering models are strong natural baselines inspired by recent work, it may not have been expressed thoroughly enough in the previous version. We updated the paper to better present these models as a baseline rather than the main proposed method (please see the answer to Weakness 1 for Reviewer Kczf). Contrary to previous work, our focus is on systematically evaluating the effect of NLI models to make models robust to irrelevant context, including in multi-hop tasks.
>
> We cite relevant recent relevant work [1,2] when presenting our filtering modules in Section 2.1, and will be happy to reference additional related work in future versions.
>
> [1] TRUE: Re-evaluating factual consistency evaluation., Honovich et al., ACL 2022
>
> [2] Attributed Question Answering: Evaluation and Modeling for Attributed Large Language Models, Bohnet et al., 2022
>
> >W2: The experimental study can be improved (please refer to my posted questions).
>
> Thank you for this comment. We responded in detail to each question and updated the paper accordingly. Specifically, we added results about generalization of our trained models in A.3 and updated  the statement about the importance of irrelevant context during fine-tuning in the Introduction.
>
> >Q1: How generalizable is the fine-tuned model? If it is applied to other QA datasets, especially with different kind of context noise, what will happen?
>
> Thank you for this interesting question. Having a domain-general model that is robust to irrelevant context is an important and interesting direction for future work as finetuning is known to potentially reduce transfer to other tasks.
>
> As a first step, we tried to evaluate the generalizability of our single-hop QA model to multi-hop questions by evaluating our NQ trained models on 2Wikihop and StrategyQA. In preliminary experiments, these models failed to generalize and generated single-hop answers even when few-shot prompted for the multi-hop tasks, suggesting they specialize in single-hop QA.
>
> Next, we tested the cross-domain generalization of multi-hop QA models. This was done by training a model on a combination of our multi-hop 2Wikihop and StrategyQA data and evaluating on the Bamboogle and Fermi datasets. We provide details for this experiment and the full results in A.3. We observe a few interesting trends: (a) our RetRobust models outperform all of the trained OOD models and are more robust to irrelevant context, and (b) in some cases, the in-context models (which are prompted in-distribution) outperform the trained models (that are evaluated out of distribution). Overall, we are hopeful that these results could help research towards a general RALM that is robust to irrelevant context, as we discuss in our updated Conclusion.
>
> >Q2: Which part of the experiment can support the statement "models finetuned solely with relevant contexts are far less robust than those finetuned with a mixture of relevant and irrelevant contexts"?
>
> Thank you for this question. This quote is based on our ablation study (Section 4), that shows that SA-RetRobust (which was trained on a mixture that ensures relevant and irrelevant contexts) consistently outperforms SA-R@1 (trained only with top-1 results from Google Search) by +0.2, +0.4, and +3.2 points when retrieving top-1, top-10, and random contexts, respectively. We modified the quote in the updated version to better highlight this result.

---

> > ### Author Response · Authors · 2023-11-22
> > **Response**
> >
> > Dear reviewer,
> >
> > We wanted to thank you again for your review, which was very helpful in improving our work. If possible, we would be very happy to know that all your concerns were addressed in the updated version and to answer any follow-up questions during the time remaining for the discussion period.

---

### Official Review · Reviewer_jc66 · 2023-10-30

**Soundness:** 3 good
**Presentation:** 3 good
**Contribution:** 3 good
**Rating:** 8
**Confidence:** 3

**Summary:**

Retrieval-augmented LM are of major interest in both applied and research contexts. This paper addresses the problem of cases where the retrieved context is not actually relevant and actually degrades task performance. The paper provides error analysis of this phenomenon over benchmark datasets covering variation in problem setting, and proposes two approaches to make RALM "robust" to irrelevant context (ie, minimize performance loss). The first approach is a simpler modular "black box" solution where a separate (NLI) module evaluates the relevance of the context to reject suspected irrelevant context and prevents it from being supplied to the LM, and the second involves fine-tuning the LM to provide correct answers even when provided irrelevant context. Experiments illustrate improved performance.

**Strengths:**

The core problem of degraded RALM performance due to irrelevant context is very compelling from both practical application and general research perspectives. Also it is useful that, besides providing a "baseline" of sorts, the simpler NLI approach is suitable for applications where fine-tuning is not feasible or greater system modularity is desired for architectural reasons.

Overall, the presentation and writing are clear.

The variation in the benchmark types (single-hop, explicit and implicit multi-hop) provided good coverage of the underlying problem and demonstrated interesting behaviors, as did the various ablations and alternative configurations.

Section 5 (Analysis) has multiple examples of the authors _manually analyzing_ examples that meet certain criteria in order to develop better insights into the behavior - this is fantastic to see and I thought it had good pay offs in terms of deeper / richer understanding. I thought the error analyses here were some of the most interesting and compelling parts of the paper.

**Weaknesses:**

I would have found it helpful to have other overall findings briefly summarized at a bit higher level for, eg a practitioner trying to build an RALM application. Something like: "NLI-filtering can increase robustness to noisy IR, but at the cost of leaving IR gains on the table in some cases due to False Negatives. If possible for your setting, fine-tuning the model with intentionally varied IR quality seems to improve robustness without sacrificing performance.""

Figure 4 and Figure 5 conveyed all the results, but I still found it a little confusing or tedious to match up the pairwise analyses from the text to the "shapes" of the bar plots. Unfortunately I don't have a specific suggestion in mind here, but it did feel like a lot of cognitive load on the reader to swivel back and forth.

The claim that the fine-tuning teaches the model _when_ to use the context is not clearly established. That is, the results show that fine-tuning in the presence of both relevant and noisy contexts improves generalization results, but the mechanism by which it accomplishes this is not really known or demonstrated.

Results with Llama-2-70B: "suggesting it has more parametric knowledge", I'm not sure this assertion is necessarily supported by the observations either

**Questions:**

Will the released dataset contain the Google search results? If not, it may be difficult to fully reproduce the results. This would also handle the fact that these queries were issued against a particular point-in-time snapshot of Wikipedia.

Minor comments / typos:
* "parametric memory" - I understand this is an increasingly common term/phrase, but the first time this term is used in the paper it might be helpful to define it.
* "in-context learning has a negative affect" - should be "effect".
* Figure 5: "impr oved" typo, should be "improved".
* "Overall, this suggest" - should be "suggests".

---

> ### Author Response · Authors · 2023-11-18
> **Response to Reviewer jc66**
>
> >W1: I would have found it helpful to have other overall findings briefly summarized at a bit higher level for, eg a practitioner trying to build an RALM application. Something like: "NLI-filtering can increase robustness to noisy IR, but at the cost of leaving IR gains on the table in some cases due to False Negatives. If possible for your setting, fine-tuning the model with intentionally varied IR quality seems to improve robustness without sacrificing performance.""
>
> Thank you for this suggestion. We updated our Conclusion with a more high-level summary.
>
> >W2: Figure 4 and Figure 5 conveyed all the results, but I still found it a little confusing or tedious to match up the pairwise analyses from the text to the "shapes" of the bar plots. Unfortunately I don't have a specific suggestion in mind here, but it did feel like a lot of cognitive load on the reader to swivel back and forth.
>
> Thank you for this helpful comment. We made several major changes to improve figures 4 and 5 and improve the readability of Section 4. For more details, please see our response to Reviewer Kczf, Weakness 2. We would be happy to get additional suggestions for improving clarity.
>
> >W3: The claim that the fine-tuning teaches the model when to use the context is not clearly established. That is, the results show that fine-tuning in the presence of both relevant and noisy contexts improves generalization results, but the mechanism by which it accomplishes this is not really known or demonstrated.
>
> Thank you for this comment. Could you kindly point us to the exact location in the paper where this claim is made? At the end of the introduction, we claim that “training LLMs when to use retrieval helps make models robust to irrelevant context and improve their overall performance.” This quote is based on results in Section 4 showing that our trained models are more robust to retrieval of irrelevant context and outperform all other baselines.
>
> >W4: Results with Llama-2-70B: "suggesting it has more parametric knowledge", I'm not sure this assertion is necessarily supported by the observations either
>
> Thank you for this comment. This quote is based on the observation that for NQ in a no retrieval setting, our in-context Llama-70B model outperforms the trained and in-context Llama-13B by 4.3 points (38.4 vs 34.1) and 8.8 points (38.4 vs 29.6), respectively (Tables 2, 3, and 6). However, we removed this quote from the updated version.
>
> >Q1: Will the released dataset contain the Google search results? If not, it may be difficult to fully reproduce the results. This would also handle the fact that these queries were issued against a particular point-in-time snapshot of Wikipedia.
>
> Thank you for this question. We cached all Google Search results and plan to release them with the final version, together with our code and models.
>
> >Q2: "parametric memory" - I understand this is an increasingly common term/phrase, but the first time this term is used in the paper it might be helpful to define it.
>
> We added a definition for parametric memory in Section 3.2.2.
>
> >Q3: typos
>
> Thank you for pointing us to these typos. We fixed them in the updated version.

---

> > ### Comment · Reviewer_jc66 · 2023-11-22
> > **Author response**
> >
> > Thank you for the thoughtful response. The plots are indeed a bit easier to follow now.
> >
> > My mechanism Q was indeed about the statement “ training LLMs when to use retrieval” - my reading was that somehow you were making a more granular claim about _how_ the fine tuned model was being made more robust: eg, like you had attention residuals vs that demonstrated the fine tuned model was not having attention to the irrelevant context tokens. The performance results do show that the fine tuned model has improved performance in the presence of irrelevant context, but it is known or shown exactly why or how this is the case.

---

> > > ### Author Response · Authors · 2023-11-23
> > > **Response**
> > >
> > > Thank you for your response and for appreciating the modifications in the new version. We agree that understanding the intrinsic mechanism by which fine-tuned LLMs learn robustness to irrelevant context is an important problem. Nevertheless, we view this as an interesting topic to explore in future work.

---

### Official Review · Reviewer_CyE8 · 2023-11-02

**Soundness:** 3 good
**Presentation:** 3 good
**Contribution:** 3 good
**Rating:** 6
**Confidence:** 4

**Summary:**

This work presents studies cases where noisy retrieval reduces accuracy in retrieval-augmented LM systems. They propose two methods. In the first one, they use off-the-shelf entailment (NLI) models to fall-back to the LM's internal knowledge when NLI models judge the retrieved context as irrelevant. This method shows some promise but is too aggressive at skipping retrieval.

In their second method, they fine-tune the LM itself so it's robust to noisy contexts in single- and multi-hop settings. While this is easy to handle for single-hop questions, the authors propose a data generation algorithm that creates fine-tuning data for multi-hop robustness. This method prompts an LLM to generate multiple decompositions of multi-hop questions, and use a self-consistency check to identify high-quality training examples.

They conduct a rich analysis that shows that irrelevant context "causes a wide range of errors, which include copying irrelevant answers from the retrieved sentences and hallucinating incorrect answers and decompositions." Their evaluation shows gains in practice across several QA benchmarks.

**Strengths:**

1. The paper is well-written and very easy to follow. (I give 3/4 for presentation only because of note #2 in weaknesses.)

2. The work is highly systematic, starting from first principles and building multiple rich systems for RALM, with well-conducted experiments sprinkled throughout to support all key claims. The results are solid.

3. The multi-hop data generation approach is novel and interesting.

**Weaknesses:**

1. If I understand correctly, you use the irrelevant context (e.g., in the single-hop case) to train the LM to answer the question by ignoring the context. Isn't this (almost) the definition of hallucination? The resulting LM will produce information not grounded in any passages. Isn't it better to abstain / request a new query, if the context is irrelevant?

2. More fundamentally, it seems like the take-away message is almost presented as "you should finetune on some examples with irrelevant/distracting context mixed in", which however is a very old message incorporated already in multiple mainstream RALM research papers from 2020 on Open-QA (if not indirectly from circa 2018 with HotPotQA, I can't confirm this one).

It seems that the bigger contribution is: how to apply this for multi-hop tasks (interesting pipeline with code-davinci-002) and the analysis conducted, though I think this demands some changes to the discussion to make it clear what precisely is portrayed as new.

**Questions:**

1. Llama2 here refers to the vanilla or the chat variant?

2. The choice of NLI model is quite underwhelming. Do you expect this to be different with good prompting of good LMs? Or with better finetuning for NLI?

3. The focus on top-1 passage weakens the search space of the ideas in the work. Have you considered filtering passages (out of several passages) with NLI, or keeping the best-scoring NLI passage?

4. The analysis conducted ends in two rich notes that I expect to yield a lot of value for this paper. I'd be willing to update my score (up or down) depending on that.

Quote 1: "In addition, the SA-R@1 that contains the top-1 results is not the best performing even when retrieving top-1 results at inference time, and is the worst performing when retrieving noisy contexts at inference time, suggesting that showing examples for retrieval during in-context learning has a negative affect that causes over-utilization of irrelevant context"

Quote 2: "for at least 36% of the cases the generated answer or decomposition is correct, but the retrieved context does not directly entail the generation. This can be partially explained by the ability of the model to combine retrieved evidence and its parametric knowledge."

How robust is Quote 1 across LLMs and selection of examples? How can Quote 2 inspire a better way to incorporate NLI?

---

> ### Author Response · Authors · 2023-11-18
> **Response to Reviewer CyE8 (1/2)**
>
> We thank the reviewer for the helpful comments. We reply in different comments due to the maximum characters limitation.
>
> >W1: If I understand correctly, you use the irrelevant context (e.g., in the single-hop case) to train the LM to answer the question by ignoring the context. Isn't this (almost) the definition of hallucination? The resulting LM will produce information not grounded in any passages. Isn't it better to abstain / request a new query, if the context is irrelevant?
>
> Thanks for this insightful question. Our goal is to train models capable of both using retrieved contexts, when relevant, as well as falling back to rely on their parametric knowledge when no such context is available. To avoid generating training examples that may cause the model to “hallucinate” incorrect facts we rely on a preliminary phase which uses an LLM (code-davinci-002) without any access to a retriever (we made this clearer in Footnote 5 in the updated version). By taking only examples which the LLM answers correctly, without any context, we assume that the necessary knowledge is more likely to be encoded in its parameters. Nevertheless, this does not provide a full-proof guarantee and we now explicitly address this in Footnote 5.
>
> Training models to abstain or request a new query given an irrelevant context is an interesting idea. When a context is classified as relevant this might improve performance, similarly to the high delta in accuracy in cases that our NLI models give a high entailment score (Table 8). However, the downside may be that models would learn to avoid using their parametric knowledge, thereby limiting their overall performance. Having models that are able to revert to their parametric when a retrieved context is irrelevant without hallucinating can be an interesting direction for future work.
>
> >W2: More fundamentally, it seems like the take-away message is almost presented as "you should finetune on some examples with irrelevant/distracting context mixed in", which however is a very old message incorporated already in multiple mainstream RALM research papers from 2020 on Open-QA (if not indirectly from circa 2018 with HotPotQA, I can't confirm this one). It seems that the bigger contribution is: how to apply this for multi-hop tasks (interesting pipeline with code-davinci-002) and the analysis conducted, though I think this demands some changes to the discussion to make it clear what precisely is portrayed as new.
>
> Thank you for this comment. It has indeed been shown that learning from examples with irrelevant context can be used to increase model performance. We discuss this in our Related Work section, e.g [1,2]. Nevertheless, we believe this observation requires revisiting in the context of the recently popularized In-Context-RALMs [3], LLMs that contain large amounts of information in their parameters and are able to answer many questions without retrieval. We believe that this phenomenon is especially relevant in datasets where multiple evidence is necessary. Some evidence may be easy to retrieve, but others, such as commonsense reasoning, will scarcely yield relevant contexts. This warrants the RALM to combine its relevant contexts with its parametric knowledge.
>
> We also agree that our automatic data generation pipeline for multi-hop questions is an important contribution. As suggested, we made several changes to the Abstract, Introduction, and Conclusion in our new version to better emphasize this point.
>
> [1] Large language models with controllable working memory, Li et al., Findings of ACL 2023
>
> [2] Large language models can be easily distracted by irrelevant context, Shi et al., ICML 2023
>
> [3] In-Context Retrieval-Augmented Language Models, Ram et al., TACL 2023
>
> >Q1: Llama2 here refers to the vanilla or the chat variant?
>
> Llama2 refers to the vanilla variant. We added this information in A.1 in the updated version.
>
> >Q2: The choice of NLI model is quite underwhelming. Do you expect this to be different with good prompting of good LMs? Or with better finetuning for NLI?
>
> Thank you for this question. Our NLI model is widely used (over 3M monthly downloads, https://huggingface.co/facebook/bart-large-mnli), trained on the Multi-NLI dataset [1], and has been shown to perform quite well in recent work [2]. As a side note, in our analysis in Section 5 we found that in more than a third of the failed examples the failure was not due to the NLI model, hence further improving the NLI model will not mitigate these issues (we discuss possible future solutions in the answer to Q4.2). However, we will also be happy to include results with a stronger NLI model in our camera-ready version. In particular, we welcome any suggestions to a particular NLI model which may be considered a better fit.
>
> [1] A Broad-Coverage Challenge Corpus for Sentence Understanding through Inference, Williams et al, ACL 2018
>
> [2] Zero-Shot Aspect-Based Sentiment Analysis, Shu et al., 2020

---

> ### Author Response · Authors · 2023-11-18
> **Response to Reviewer CyE8 (2/2)**
>
> >Q3: The focus on top-1 passage weakens the search space of the ideas in the work. Have you considered filtering passages (out of several passages) with NLI, or keeping the best-scoring NLI passage?
>
> Indeed, recent work has shown that NLI models can be used for re-ranking in order to increase performance and attribution [1]. In our work, we focus on how LLMs utilize retrieved contexts, rather than on the re-ranking of retrieved contexts. Hence, we use the (very) strong Google Search as our retriever, which we assume contains a built-in re-ranker and will be hard to outperform simply with NLI re-ranking. Nevertheless, we will be happy to include results with this baseline (Google top-k retrieval + NLI re-ranking) in future versions.
>
> [1] Attributed Question Answering: Evaluation and Modeling for Attributed Large Language Models, Bohnet et al., 2022
>
> >Q4.1: How robust is Quote 1 across LLMs and selection of examples?
>
> Thank you for these interesting questions. We see these trends for both the 13B and 70B Llama-2 models and across multiple evaluation datasets (Tables 2 and 6).
>
> For the 13B model, when averaging across datasets, we observe that  the model prompted with top-1 results (R@1) underperforms the in-context no-retrieval baseline (NoRet) by more than 2 points (Table 2). For the 70B model, R@1 has no advantage and all models perform relatively similarly (Table 6).
>
> When retrieving low-ranked or random results, the R@1 model has the worst performance. For example, for the 13B model (Table 2) performs worse than the NoRet prompt by 3.1 and 9.8 points respectively on average across all datasets and by 8.6 and 4.8 points for NQ on the 70B model (Table 6). Thus, the trend where showing examples for retrieval during in-context learning has a negative effect that causes over-utilization of irrelevant context is consistent across datasets and both model sizes. We updated the quote in Section 4 to better express these points.
>
> >Q4.2: How can Quote 2 inspire a better way to incorporate NLI?
>
> We are indeed hopeful our analysis can inspire future work. For example, one can examine the effect retrieval augmentation has on generation probability or on the model’s internal knowledge base [1]. Other options can focus on the NLI side, for example by training the NLI model to generate a justification to its prediction which can be further evaluated against a model’s parametric knowledge. Following this advice we updated the relevant quote in our analysis section, adding potential ideas for future work.
>
> [1] Crawling The Internal Knowledge-Base of Language Models, Cohen et al., Findings of EACL 2023

---

> > ### Author Response · Authors · 2023-11-22
> > **Response**
> >
> > Dear reviewer,
> >
> > We wanted to thank you again for your review, which was very helpful in improving our work. If possible, we would be very happy to know that all your concerns were addressed in the updated version and to answer any follow-up questions during the time remaining for the discussion period.

---

> > > ### Comment · Reviewer_CyE8 · 2023-11-22
> > >
> > > Thank you for the detailed responses. I have read them and I'm considering whether to adjust my scores. As of now, I will keep my scores the same.

---

### Author Response · Authors · 2023-11-18
**Response to all Reviewers**

We thank the reviewers for their helpful reviews. We added a new version of the paper that addresses the comments made by the reviewers. More specifically, our main changes include:
- Rewriting of Section 4 and updating Figures 4 and 5 in line with the comments made by Reviewer jc66 and Reviewer Kczf.
- Updating the discussion based on the comments of Reviewer CyE8 and Reviewer jc66.
- New experimental results on the out-of-domain generalization of the fine-tuned LLM (following the comment by reviewer dWp6).
- Addressing the comments by Reviewers dWp6 and Kczf regarding the NLI model, and its framing as a baseline to our models, rather than a main contribution.
- An updated analysis section, with statistical significance tests and future application of our methods (addressing comments by Reviewer Kczf and Reviewer CyE8).

We address each comment in our individual response to the reviewers.

---

### Public Comment · ~JK1 · 2024-08-09
**Typo in abstract?**

There are two 'are' in the first sentence of the abstract:
- Retrieval-augmented language models (RALMs) hold promise to produce language understanding systems that ***are are factual***, efficient, and up-to-date.

---

### Meta-Review · Area_Chair_KRSt · 2023-12-13

**Metareview:**

This paper addresses the challenge of retrieved context being irrelevant, which can degrade task performance. Through error analysis across benchmark datasets, the paper introduces a modular "black box" solution, where a separate module evaluates context relevance, and a fine-tuning approach, both resulting in improved performance. I enjoyed reading the paper and all the reviews are also positive.

**Justification For Why Not Higher Score:**

NA

**Justification For Why Not Lower Score:**

NA

---

### Decision · Program_Chairs · 2024-01-16

Accept (poster)